# Alpha-Tocotrienol Prevents Oxidative Stress-Mediated Post-Translational Cleavage of Bcl-xL in Primary Hippocampal Neurons

**DOI:** 10.3390/ijms21010220

**Published:** 2019-12-28

**Authors:** Han-A Park, Nelli Mnatsakanyan, Katheryn Broman, Abigail U. Davis, Jordan May, Pawel Licznerski, Kristi M. Crowe-White, Kimberly H. Lackey, Elizabeth A. Jonas

**Affiliations:** 1Department of Human Nutrition and Hospitality Management, College of Human Environmental Sciences, The University of Alabama, Tuscaloosa, AL 35487, USA; kabroman@crimson.ua.edu (K.B.); audavis@crimson.ua.edu (A.U.D.); jmmay4@crimson.ua.edu (J.M.);; 2Department of Internal Medicine, Section of Endocrinology, Yale University, New Haven, CT 06511, USA; nelli.mnatsakanyan@yale.edu (N.M.); pawel.licznerski@yale.edu (P.L.); elizabeth.jonas@yale.edu (E.A.J.); 3Department of Biological Sciences, College of Arts and Sciences, The University of Alabama, Tuscaloosa, AL 35487, USA; lacke003@ua.edu

**Keywords:** Bcl-xL, ∆N-Bcl-xL, antioxidant, mitochondria, tocotrienol

## Abstract

B-cell lymphoma-extra large (Bcl-xL) is an anti-apoptotic member of the Bcl2 family of proteins, which supports neurite outgrowth and neurotransmission by improving mitochondrial function. During excitotoxic stimulation, however, Bcl-xL undergoes post-translational cleavage to ∆N-Bcl-xL, and accumulation of ∆N-Bcl-xL causes mitochondrial dysfunction and neuronal death. In this study, we hypothesized that the generation of reactive oxygen species (ROS) during excitotoxicity leads to formation of ∆N-Bcl-xL. We further proposed that the application of an antioxidant with neuroprotective properties such as α-tocotrienol (TCT) will prevent ∆N-Bcl-xL-induced mitochondrial dysfunction via its antioxidant properties. Primary hippocampal neurons were treated with α-TCT, glutamate, or a combination of both. Glutamate challenge significantly increased cytosolic and mitochondrial ROS and ∆N-Bcl-xL levels. ∆N-Bcl-xL accumulation was accompanied by intracellular ATP depletion, loss of mitochondrial membrane potential, and cell death. α-TCT prevented loss of mitochondrial membrane potential in hippocampal neurons overexpressing ∆N-Bcl-xL, suggesting that ∆N-Bcl-xL caused the loss of mitochondrial function under excitotoxic conditions. Our data suggest that production of ROS is an important cause of ∆N-Bcl-xL formation and that preventing ROS production may be an effective strategy to prevent ∆N-Bcl-xL-mediated mitochondrial dysfunction and thus promote neuronal survival.

## 1. Introduction

B-cell lymphoma—extra large (Bcl-xL) protein is an anti-apoptotic member of the Bcl2 family, containing bcl-2 homology (BH)1, BH2, BH3, and BH4 domains. The pro-survival function of Bcl-xL has been well documented in various tissues and cells, including the central nervous system (CNS). A primary function of Bcl-xL is to inhibit pro-apoptotic members of the Bcl2 family and to prevent the intrinsic death pathway [1]. For example, Bax and Bak, multi-domain effectors without a BH4 domain, undergo oligomerization and form pores in mitochondrial membranes that release cytochrome c. This event promotes formation of the apoptosome, which causes neuronal death. Bcl-xL binds directly to pro-death Bax and Bak to prevent mitochondrially mediated apoptotic pathways, and also binds to BH3-only proteins that regulate Bax/Bak oligomerization [2,3,4,5,6,7,8]. In addition to its anti-apoptotic role, Bcl-xL supports functions of the brain during physiological and pathological processes. Bcl-xL blocks abnormal mitochondrial channel activity and enhances ATP production [9,10,11,12]. The improved mitochondrial metabolism supports energy-requiring events in the brain such as neurite outgrowth [13,14] and neurotransmission [15,16,17,18,19]. Approaches that prevent Bcl-xL loss in both in vitro and in vivo systems have been reported to protect the brain against pathological events [20,21,22].

In contrast to its pro-survival or neuroprotective functions, Bcl-xL also assists in pro-death pathways. During excitotoxic challenge such as cerebral ischemia, Bcl-xL undergoes caspase-dependent N-terminal truncation and is converted to a cleaved form, ∆N-Bcl-xL [22,23,24,25,26,27,28,29]. Caspase 3 is a central player that cleaves at the Asp61 site of full-length Bcl-xL; application of caspase inhibitors prevents formation of ∆N-Bcl-xL [22,23,24,25,26,28]. Accumulation of ∆N-Bcl-xL is responsible for mitochondrial dysfunctions such as large conductance mitochondrial channel activity [28,30], and reduction of mitochondrial membrane potential associated with neuronal death [26]. Therefore, strategies designed to block ∆N-Bcl-xL production or to inactivate pro-death functions of ∆N-Bcl-xL may be important in eliciting neuroprotective mechanisms.

Alpha-tocotrienol (α-TCT), a member of the vitamin E family, is a lipid-soluble, chain-breaking antioxidant. Studies have reported that α-TCT has stronger antioxidant properties than other vitamin E analogs [31,32,33], and exhibits superior ability to scavenge the free radicals that cause lipid peroxidation [34]. α-TCT efficiently penetrates across lipid bilayers and the blood–brain barrier, and it is successfully delivered to the brain [35,36,37]. Indeed, neuroprotective properties of α-TCT have been reported in various CNS-associated disease models. Orally supplemented α-TCT protects rodent and canine brain against ischemic stroke [37,38,39,40]. α-TCT also showed beneficial effects in human subjects with a high risk of stroke [41]. In addition to stroke models, decreased levels of plasma α-TCT are associated with Alzheimer’s disease and mild cognitive impairment among older adults [42]. Long-term supplementation with a tocotrienol-rich fraction (TRF) to AβPP/PS1 transgenic mice expressing amyloid protein precursor attenuated amyloid beta deposition in the brain and improved cognition [43,44,45]. Aged rats supplemented with TRF also showed improved spatial learning and memory and reduced anxiety levels [46].

In this study, we found that oxidative stress during excitotoxicity induces cleavage of Bcl-xL to form ∆N-Bcl-xL. This might be the key modification that triggers mitochondrially associated neuronal death under these conditions. We found that application of the antioxidant α-TCT improves mitochondrial function by inhibiting ∆N-Bcl-xL formation in mitochondria. Mitochondrially localized ∆N-Bcl-xL is therefore a novel target of α-TCT-mediated neuroprotection.

## 2. Results

### 2.1. α-TCT Protects Primary Hippocampal Neurons against Excitotoxicity

Glutamate is an abundantly expressed neurotransmitter and plays an important role in learning and memory. However, during pathological processes such as stroke, brain cells are exposed to a surge of glutamate, and elevated glutamate triggers a series of molecular events that cause reactive oxygen species (ROS) production and mitochondrial dysfunction, resulting in neuronal death (excitotoxicity) [37,47,48]. We have previously shown dose-dependent glutamate toxicity in primary hippocampal neurons [26]. We chose to apply 20 μM glutamate to induce moderate levels of neurotoxicity in this study. To test the function of α-TCT against glutamate excitotoxic challenge, we measured the activity of lactate dehydrogenase (LDH), a marker for loss of cell membrane integrity. Based on our pilot screening, we chose 1 μM α-TCT as a functional concentration to exert neuroprotective properties against excitotoxicity in our in vitro culture system. Primary hippocampal neurons were treated with α-TCT, glutamate, or a combination of both. The vehicle control group was treated with ethanol and PBS. After 24 h incubation, α-TCT treatment alone did not alter LDH leakage from primary hippocampal neurons, but α-TCT prevented glutamate-induced LDH leakage (Figure 1A). Similarly, glutamate treatment significantly increased propidium iodide (PI) uptake, a marker for late apoptotic and necrotic death, and α-TCT co-treatment reduced glutamate-induced PI staining (Figure 1B,D). Fluorescent calcein retention indicates cell survival. Glutamate treatment significantly lowered the calcein fluorescent intensity of viable cells, but α-TCT co-treatment augmented calcein intensity in primary hippocampal neurons (Figure 1C,E). Our data consistently showed that glutamate-induced excitotoxicity increases hippocampal death, whereas application of α-TCT protects neurons against excitotoxic stimulation.

### 2.2. α-TCT Attenuates Glutamate-Induced Oxidative Stress in the Mitochondria.

We performed an oxygen radical absorbance capacity (ORAC) assay in primary hippocampal neurons by quantifying the loss of fluorescein fluorescence via presence of peroxyl radicals. We found that excitotoxicity impaired clearance of peroxyl radicals in 7% randomly methylated beta-cyclodextrin (RMCD) buffer, indicating vulnerability of neurons to oxidative stress and the need for lipophilic antioxidant support (Figure 2A). In order to test whether a lipophilic antioxidant, α-TCT, could play a role in the prevention of glutamate-induced oxidative stress in hippocampal neurons, we measured intracellular hydrogen peroxide levels using 2′,7′-dichlorofluorescein (DCF). During preliminary screening, we found that 24 h glutamate treatment caused failure of DCF retention in hippocampal neurons due to loss of the neuronal population. In order to eliminate data influenced by neuronal death, we performed ROS studies at 6 h after treatments, where there was no appreciable death. Primary hippocampal neurons treated with glutamate for 6 h had significantly increased DCF fluorescent intensity. However, α-TCT treated neurons showed decreased DCF level during glutamate challenge (Figure 2B). Next, we tested whether α-TCT prevents superoxide accumulation in the mitochondria. Primary hippocampal neurons were treated with α-TCT, glutamate, or a combination of both for 6 h, then neurons were stained with mitoSOX, a fluorescent dye for detecting mitochondrial superoxide. Glutamate challenge significantly increased the mitoSOX positive signal, indicating accumulation of mitochondrial ROS, while α-TCT attenuated the fluorescence intensity of mitoSOX (Figure 2C,D). Our data showed that application of the antioxidant, α-TCT during early excitotoxic insult attenuates generation of oxidative stress and prevents ROS-induced neuronal death signaling.

### 2.3. α-TCT Decreases Mitochondrial Formation of ∆N-Bcl-xL in Primary Hippocampal Neurons

Although full length Bcl-xL is required for normal mitochondrial function and hippocampal survival, accumulation of ∆N-Bcl-xL, the N-terminal cleavage product of Bcl-xL, is causative in promoting hippocampal death during brain ischemia [22,23,26]. We have recently reported that glutamate-induced excitotoxicity is also responsible for ∆N-Bcl-xL formation, and we found that ∆N-Bcl-xL protein was detected after 6 h glutamate treatment in primary hippocampal neurons [26]. Caspases, in particular caspase 3, are reported to cleave the aspartic acid peptide bond of Bcl-xL to form ∆N61 Bcl-xL [24,25,49]. Application of caspase inhibitors such as Ac-DEVD-CHO and zVAD-fmk blocks formation of ∆N-Bcl-xL [26,28,50]. However, it is still unclear if ROS-induced hippocampal loss is associated with caspase 3-dependent ∆N-Bcl-xL formation.

To test whether ∆N-Bcl-xL formation during excitotoxity is due to ROS production, and whether treatment with the antioxidant (α-TCT) would protect neurons via inhibiting formation of ∆N-Bcl-xL, primary hippocampal neurons were treated with 1 μM α-TCT, 20 μM glutamate, or a combination of both for 24 h, and then the protein levels of ∆N-Bcl-xL, Bcl-xL, and active caspase 3 (Figure 3A–D) were quantified. We have previously reported that the anti-∆N-Bcl-xL antibody is highly selective against ∆N-Bcl-xL protein in hippocampal neurons and binds less effectively to full length Bcl-xL [26]. Glutamate significantly increased ∆N-Bcl-xL (Figure 3A,B) without changing the abundance of full length Bcl-xL (Figure 3A,C). Neurons treated with glutamate showed increased levels of active caspase 3 protein (Figure 3A,D). This suggests that caspase 3-mediated ∆N-Bcl-xL production is a key mechanism contributing to ∆N-Bcl-xL accumulation during excitotoxicity. α-TCT treatment attenuated glutamate-induced ∆N-Bcl-xL formation and inhibited accumulation of active caspase 3 (Figure 3A–D), suggesting that ROS was responsible for activating the pathway. We further tested the activity of caspase 3 by measuring fluorescent 7-amino-4-methylcoumarin (AMC), the product of active caspase 3. Glutamate-mediated excitotoxicity significantly increased caspase 3 activity after 6 h incubation, which is consistent with the time-course of ∆N-Bcl-xL formation in our previous study [26]. Treatment with α-TCT prevented glutamate-induced caspase 3 activity in primary hippocampal neurons (Figure 3E).

### 2.4. α-TCT Prevents ∆N-Bcl-xL-Induced Mitochondrial Dysfunction.

To test whether α-TCT-mediated neuroprotection acts via regulation of mitochondrial function, we measured neuronal mitochondrial membrane potential using tetramethylrhodamine (TMRM). Cationic TMRM accumulates in negatively charged active mitochondria. Primary hippocampal neurons were treated with α-TCT, glutamate, or a combination of both for 6 h. TMRM fluorescence was significantly decreased in glutamate-treated neurons, indicating a loss of mitochondrial membrane potential. Although α-TCT alone did not affect mitochondrial membrane potential compared to vehicle-treated neurons, co-treatment of α-TCT and glutamate significantly attenuated glutamate-induced loss of mitochondrial membrane potential (Figure 4A,B). We further examined the metabolic status of each group by measuring ATP production. Glutamate-treated neurons had significantly decreased ATP levels, whereas α-TCT prevented glutamate-induced ATP loss (Figure 4C). To determine whether glutamate-induced mitochondrial dysfunction is associated with caspase 3-mediated ∆N-Bcl-xL formation, we treated primary neurons with the caspase 3 inhibitor Ac-DEVD-CHO (10 μM). This concentration has been previously shown to inhibit production of ∆N-Bcl-xL [26]. Treatment with Ac-DEVD-CHO prevented glutamate-mediated loss of mitochondrial membrane potential (Figure 4D and Appendix A). We had previously determined that ∆N-Bcl-xL promoted loss of mitochondrial membrane potential and could do so by its direct interaction with mitochondria [26]. In that report, we showed that Bax was activated upon ∆N-Bcl-xL formation and that Bax activation was prevented by pharmacological inhibition of ∆N-Bcl-xL formation, but we did not demonstrate a direct interaction between the two molecules. Here, in order to determine if ∆N-Bcl-xL binds directly to Bax, we overexpressed then immunoprecipitated HA-tagged ∆N-Bcl-xL in HEK293T cells. In the absence of α-TCT, Bax was found in complex with ∆N-Bcl-xL, but this interaction was prevented by α-TCT exposure (Figure 5A). This result suggests that α-TCT prevents the interaction between ∆N-Bcl-xL and active monomeric Bax. The input demonstrates the presence of monomeric Bax in the total cell lysate.

If α-TCT prevents the interaction between the two molecules, it may prevent a direct effect of ∆N-Bcl-xL on loss of mitochondrial membrane potential. We therefore tested whether α-TCT can protect mitochondria when ∆N-Bcl-xL is already formed. We reported previously that hippocampal neurons overexpressing ∆N-Bcl-xL showed impairment of mitochondrial potential in the absence of exogenous glutamate [26]. Here, we treated ∆N-Bcl-xL-overexpressing neurons with or without α-TCT, then measured the intensity of TMRM. Neurons treated with α-TCT exhibited attenuated ∆N-Bcl-xL-mediated loss of mitochondrial potential (Figure 5B,C) indicating a protective role of α-TCT in the presence of ∆N-Bcl-xL. Taken together, our data suggest that α-TCT protects neurons indirectly, by reducing formation of ∆N-Bcl-xL, and directly, by preventing interaction of ∆N-Bcl-xL with partners at the mitochondrial membrane. One such partner is clearly Bax, but another might be a constitutive component of the inner membrane that actively participates in mitochondrial depolarization, such as the permeability transition pore (mPTP).

## 3. Discussion

In this study, we found that production of ΔN-Bcl-xL is an important cellular event during excitotoxicity-mediated ROS production. Excitotoxic stimulation produces mitochondrial ROS which activates caspase 3, cleaving Bcl-xL (Figure 3), and accumulation of ∆N-Bcl-xL is highly associated with neuronal death. Approaches that inhibit ∆N-Bcl-xL formation have been shown to be neuroprotective in various models. For example, treatment with ABT-737, a pharmacological inhibitor that blocks ∆N-Bcl-xL accumulation, protects hippocampal neurons in vivo and in vitro against ischemia and excitotoxicity [22,26,29]. Induction of ischemic stroke in genetically modified Bcl-xL-cleavage-resistant animals prevents the appearance of large conductance mitochondrial ion channel activity [22]. PINK1-mediated phosphorylation of serine 62 residue (on Bcl-xL) prevents truncation of Bcl-xL, thus protecting mitochondria and SH-SY5Y cells against neurotoxicity [51].

Accumulation of ∆N-Bcl-xL in the mitochondria alters mitochondrial function and triggers mitochondria-mediated death signaling [22,23,26,27,28]. Although ∆N-Bcl-xL insertion into the mitochondrial outer membrane could eventually lead to abnormalities of inner membrane function (e.g., cytochrome c release through the outer membrane causes electron transport deficiency and activates caspases) [30,52], there is abundant evidence suggesting that expression of Bcl-xL also occurs on the inner membrane [9,26]. Thus, ∆N-Bcl-xL could be formed in the mitochondrial inner membrane, directly regulating mitochondrial inner membrane function. Various laboratories have found expression of full length Bcl-xL and other anti-apoptotic Bcl-2 family members on the mitochondrial inner membrane [9,26,53,54]. The purified mitochondrial inner membrane fraction clearly shows expression of Bcl-xL protein [10,26], and Bcl-xL interacts with ATP synthase, which localizes to the mitochondrial inner membrane and matrix [9,55]. In addition, the active form of caspase 3, the enzyme that cleaves full length Bcl-xL and forms ∆N-Bcl-xL, is expressed in mitochondria [56,57]. Thus, the mitochondrial inner membrane and matrix, particularly when it is exposed to oxidative stress, may provide a suitable environment for the production of ∆N-Bcl-xL.

Full-length Bcl-xL binds directly with the α and β subunits of ATP synthase, and this interaction enhances mitochondrial energy metabolism by closing an inner membrane leak channel [9]. Glutamate toxicity disrupts Bcl-xL and ATP synthase interaction within the matrix of mitochondria [55]. Glutamate toxicity also activates caspases which may then freely interact with Bcl-xL in the mitochondrial matrix, forming ∆N-Bcl-xL. Since the chemical structure of ∆N-Bcl-xL resembles Bcl-xL protein binding partners such as Bak and Bax [2,3,4,5,6], ∆N-Bcl-xL may bind to and sequester full length Bcl-xL, thereby inhibiting Bcl-xL and ATP synthase interaction and leading to inefficient ATP production. As shown in Figure 4, primary hippocampal neurons treated with high glutamate concentration fail to produce sufficient ATP, and we speculate that the energy deficit may cause continuous operation of the electron transport chain, which will serve as the main source of ROS [58]. Our data corroborated that the main source of ROS upon glutamate toxicity is inside the mitochondria. We suggest that ROS produced at the inner membrane electron transport chain is associated with activation of caspase 3 [59,60] and we suggest that this leads to caspase-mediated Bcl-xL cleavage. Therefore, the mitochondrial inner membrane is the crucial location for regulating ROS production, mitochondrial function, and death.

Vitamin E is well known for its ability to penetrate into or to be embedded in cellular membranes. Mitochondrial electron transport produces superoxide, which is then transformed into hydrogen peroxide and hydroxyl radicals. These ROS can directly injure the membrane structure or initiate lipid peroxidation [61]. α-TCT has been reported to show greater antioxidant activity in the phospholipid membrane environment than tocopherols (TCPs) [36], due to its unsaturated farnesyl isoprenoid tail with three double bonds. Membrane-embedded α-TCT may act as the first defense system to scavenge reactive molecules produced during excitotoxic stimulation. In our experiments, α-TCT treatment decreased ROS production, especially in the mitochondria (Figure 2), and α-TCT protected mitochondrial function (Figure 4) which eventually improved neuronal survival against the excitotoxic insult (Figure 1). In addition, we found that α-TCT treatment prevented loss of mitochondrial potential in ∆N-Bcl-xL-overexpressing neurons (Figure 5). α-TCT may prevent ∆N-Bcl-xL-mediated ROS production, thus blocking secondary ∆N-Bcl-xL production via a positive feedback loop. Although there have been limited studies elucidating the functions of α-TCT at subcellular levels, mitochondria have been suggested as an important target in vitamin-E-mediated protection. α-TCT-rich-TRF-fed rats are resistant to radiation-induced mitochondrial permeability transition pore opening and show improved mitochondrial respiration [62]. Delivery of rice bran extract containing high levels of α-TCT improves mitochondrial function, increasing mitochondrial membrane potential and ATP levels in in vivo and in vitro models [63,64]. Vitamin E regulates mitochondrial genes encoding proteins that are involved in ATP biosynthesis [65]. α-TCP supplementation inhibits alteration of age-associated ATP synthase gene expression and upregulation of caspase 3 [65]. In addition to inhibition of ∆N-Bcl-xL expression by scavenging ROS, α-TCT may interfere with transcription of caspases which may protect against mitochondrial dysfunction and neuronal death.

Although in this study, we primarily focused on investigating the mechanisms of ROS-mediated ∆N-Bcl-xL production, and a strategy to inhibit ∆N-Bcl-xL formation using antioxidant α-TCT, we speculate that α-TCT-mediated neuroprotection may not be limited to its inhibitory role in ∆N-Bcl-xL production. The homology model of ∆N-Bcl-xL reveals potential docking sites for α-TCT (Figure 6) suggesting possible formation of a ∆N-Bcl-xL–α-TCT complex after α-TCT treatment. Seven amino acid residues—Phe97, Arg100, Tyr101, Phe105, Leu130, Arg 139, and Tyr195—were identified as potential docking sites with α-TCT. Interestingly, Phe97, Tyr101, and Leu130 in Bcl-xL are important locations that interact with pro-death Bcl2 proteins such as Bax and Bak [2,66,67]. Our HEK293T cell system showed that treatment with α-TCT lowers the interaction between ∆N-Bcl-xL and Bax in vitro (Figure 5). ∆N-Bcl-xL could form a heterodimer with Bax or Bak, participating in Bax-/Bak-mediated oligomerization in the mitochondrial inner or outer membrane, since ∆N-Bcl-xL conserves BH3 docking sites. Occupation of the docking sites by application of α-TCT may prevent opening of mitochondrial pores (Figure 7). Therefore, it will be important to investigate ∆N-Bcl-xL protein binding partners to elucidate additional roles of α-TCT in regulating apoptotic pathways in future projects. Studies have also shown that Tyr101 and Phe105 residues interact with Beclin-1, a key player in autophagy [68,69]. Thus, α-TCT docking at Tyr101 and Phe105 in ∆N-Bcl-xL may influence the accessibility of Beclin-1 to sites necessary for the regulation of mitochondrial protein degradation. In addition, a pharmacological inhibitor of Bcl-xL, A-1155463, has been reported to target Phe105 in the loose alpha helix [70]. Although A-1155463 is designed to inhibit the anti-apoptotic effect of Bcl-xL, other pharmacological inhibitors of Bcl-xL, such as ABT-737 and WEHI-539, have been shown to block ∆N-Bcl-xL directly [22,26,29]; thus, A-1155463 may also exert inhibitory effects on ∆N-Bcl-xL. Since α-TCT is capable of interacting with Phe105, α-TCT potentially exhibits drug-like properties similar to A-1155463.

Taken together, this study showed that mitochondrial ∆N-Bcl-xL is an important therapeutic target during oxidative stress in mammalian neurons. We suggest a novel nutritional strategy to prevent ROS-mediated ∆N-Bcl-xL formation in the mitochondria (Figure 7). We speculate that α-TCT provides neuroprotective effects even after ∆N-Bcl-xL formation. Continued investigation of α-TCT-mediated ∆N-Bcl-xL regulation may provide further mechanisms pertaining to α-TCT neuroprotection.

## 4. Materials and Methods

### 4.1. Culture of Primary Hippocampal Neurons

Primary rat hippocampal neurons were prepared from rat feti (Sprague–Dawley, Day 18 of gestation; Harlan, Indianapolis, IN) as described previously [13,26,71,72]. Briefly, neurons (0.3 × 10^6^ cells/35 mm plate) were grown in neurobasal medium supplemented with B-27, glutamine, and antibiotics (Invitrogen Gibco Life Technologies, Carlsbad, CA, USA) for 20–22 days in vitro (DIV), and treated with glutamate 20 μM, α-TCT(1 μM), or a combination of both as described in relevant figure legends. All protocols were approved by the Institutional Animal Care Committee (IACUC) of University of Alabama, Tuscaloosa, AL (17-06-0324) and Yale University, New Haven, CT (2019-10388).

#### 4.1.1. α-tocotrienol (TCT) Treatment

A stock solution of α-TCT (Cayman chemical, Ann Arbor, MI) was prepared in ethanol and added into the cell culture medium 20 min prior to glutamate treatment. Ethanol was used as the vehicle control in all experiments with α-TCT. 

#### 4.1.2. Glutamate Treatment

20 µM of glutamate was freshly prepared in sterile PBS and added into the cell culture medium. The vehicle group for glutamate experiments was treated with sterile PBS. 

#### 4.1.3. Ac-DEVD-CHO Treatment

10 µM of Ac-DEVD-CHO was freshly prepared in sterile DMSO and added into the cell culture medium. The vehicle group was treated with sterile DMSO. 

### 4.2. Viability Assay

#### 4.2.1. Lactate Dehydrogenase (LDH) Assay

The level of cytotoxicity in primary hippocampal neurons was assayed by measuring leakage of LDH using an in vitro toxicology assay kit (Sigma-Aldrich, St louis, MO, USA) as previously described [13,26,73]. In brief, the culture media and lysed cells were collected after treatment of neurons as with α-TCT and glutamate for 24 h. The LDH assay mixture was made according to the manufacturer’s protocol and added to each sample. After 20 min incubation, the reaction was terminated by adding 1N HCl. LDH activity was spectrophotometrically measured with a VICTOR^3^ multilabel reader (PerkinElmer, Waltham, MA, USA) with absorbance set at 490 nm. Data were calculated by finding the activity of LDH leaked into the medium by damaged cells / total LDH activity in the culture. 

#### 4.2.2. Calcein-AM and Propidium Iodide (PI)

Viable or dead cells were stained with Calcein-AM or PI as previously described [13,26,74]. After treatment of neurons with α-TCT and glutamate for 24 h, 25 nM Calcein-AM or 0.5 μM PI (Invitrogen, Molecular Probes, Carlsbad, CA, USA) was added into the culture medium for 30 min at 37 °C in the dark. Images were taken using a Zeiss Axiovert 200 microscope (Zeiss, Oberkochen, Germany) using consistent exposure time. The number of PI positive neurons or calcein fluorescence densitometry per cell was analyzed using AxioVision 4.8 (Zeiss).

### 4.3. Measurement of Mitochondrial Potential (Δψ)

Mitochondrial membrane potential (Δψ) was measured using the fluorescent lipophilic cationic dye tetramethylrhodamine methyl ester (TMRM, Invitrogen, Molecular Probes, Carlsbad, CA, USA), which accumulates within mitochondria in a potential-dependent manner [73,75]. Primary hippocampal neurons were stained with 5 nM TMRM for 30 min at 37 °C in the dark. Images were taken using a Nikon C2 Laser Scanning Confocal Microscope and TMRM fluorescence densitometry was analyzed using ImageJ with FIJI plug-in.

### 4.4. Measurement of ATP Production

Primary hippocampal neurons were seeded on 96 well plates (0.015 × 10^6^ neurons/ well) for DIV 20–22. Neurons were treated with α-TCT, glutamate, or a combination of both for 8 h. Neuronal ATP production was measured using the ATPlite™ Luminescence Assay System (PerkinElmer, Waltham, MA, USA) according to the manufacturer’s protocol. In brief, the plates were washed with sterile PBS, and cells were lysed on the orbital shaker at 700 rpm for 5 min. Cells were then incubated with substrate (luciferin) on the orbital shaker at 700 rpm for 10 min. The reaction between ATP, luciferase, and luciferin produced bioluminescence. ATP-induced luminescence was measured with a VICTOR^3^ multilabel reader (PerkinElmer, Waltham, MA, USA).

### 4.5. ROS Measurement

#### 4.5.1. 2′,7′-dichlorodihydrofluorescein Diacetate (H_2_DCFDA) Staining

Primary hippocampal neurons were treated with 10 µM of DCF (Invitrogen) solution prepared in a light-protected vessel, then incubated for 30 min at 37 °C in the dark [76] and processed as per the manufacturer’s protocol. After incubation, neurons were carefully washed with pre-warmed HBSS. Intracellular fluorescence was measured using a fluorescent microplate reader (CLARIOstar, BMG Labtech) at excitation and emission wavelengths of 470-15 and 515-20 nm, respectively. 

#### 4.5.2. mitoSOX Staining

Production of mitochondrial ROS was analyzed using MitoSOX Red (Invitrogen). The MitoSOX Red dye is oxidized by superoxide in the mitochondria, emitting red fluorescence. After treatment of neurons as described in relevant figure legends, 1.25 μM of MitoSOX and 2 μg/mL 4′,6-diamidino-2-phenylindole (DAPI ) were added to the cell culture medium. Cultures were incubated for 30 min at 37 °C and washed twice with warm HBSS. Fluorescent images were taken with a Zeiss Axio Vert.A1 microscope and analyzed using AxioVision 4.9. 

#### 4.5.3. Antioxidant Capacity

Samples were deproteinated according to a published method using methanol/acetonitrile/acetone (1:1:1, *v/v/v*) added to the sample in a ratio of 1:4 (*v/v*) [77]. This method allows for detection of small molecular weight antioxidants <6 kDa in size. The antioxidant capacity of neuronal lysates was measured using the oxygen radical absorbance capacity (ORAC) assay on a FLUOstar Optima plate reader (BMG Labtech, Ortenberg, Germany) in accordance with the method by Prior et al [78]. Samples were prepared in 7% randomly methylated beta-cyclodextrin (RMCD) buffer, liberating hydrophobic antioxidants while blocking release of hydrophilic antioxidants. The compound 2,2-azobis(2-amidino-propane) dihydrochloride (AAPH) was used as the peroxyl radical generator and Trolox, a water-soluble analogue of vitamin E, was used as the reference antioxidant standard. Peroxyl radicals cause oxidation of a fluorescein probe decreasing fluorescence. Changes in the loss of fluorescein fluorescence in the RMCD buffer indicates hydrophobic antioxidant capacity.

### 4.6. Immunoblots

After treatment of α-TCT and glutamate for 24 h, the primary hippocampal neurons were scraped and lysed in the 1X cell signaling buffer (Cell signaling Technology, Danvers, MA) and protein concentration was determined using BCA protein reagents (Thermo Scientific, Rockford, IL). Samples (50–100 µg of protein/lane) were separated on a 4–12% SDS–polyacrylamide gel (Bio-Rad, Hercules, CA) and probed with anti-∆N-Bcl-xL (1:100 dilution, Aves labs, Tigard, OR), anti-∆N-Bcl-xL (1:10, Pacific Immunology, Ramona, CA), anti-active caspase 3 (Abcam, 1:100), Bcl-xL (Cell signaling, 1:1000), and anti-beta actin (sigma, 1:1000). Anti-∆N-Bcl-xL was custom-produced (peptide sequence: CZ DSP AVN GAT GHS SSL D. 1:100, Aves labs; 1:10, Pacific Immunology). Membranes were treated with ECL reagents (Perkin Elmer, Waltham, MA) and scanned with a C-DiGit blot scanner (Li-Cor, Lincoln, NE). Scanned images were analyzed using ImageJ software (National Institutes of Health, Bethesda, MD).

### 4.7. Caspase 3 Activity

Caspase 3 activity was measured by using a Caspase 3 activity assay kit according to the manufacturer’s protocol (Cell Signaling). In brief, primary hippocampal neurons were lysed and then incubated with substrate solution containing Ac-DVD-AMC at 37 °C in the dark. After 30 min, relative fluorescence unit produced by Ac-DEVD-AMC cleavage were measured at 380 nm excitation and 420–460 nm emission using CLARIOstar (BMG Labtech, Ortenberg, Germany).

### 4.8. Homology Modeling and Protein–Ligand Docking

The homology model for ΔN-Bcl-xL was built based on the crystal structure of rat Bcl-xL (PDB ID: 1AF3) [67] using the Swiss Model software (Swiss-Pdb Viewer, Lausanne, Switzerland) [79]. Mcule web server was used for docking of α-Tocotrienol into the rat ΔN-Bcl-xL homology model. UCSF Chimera (UCSF Chimera-visualization system for exploratory research and analysis) was used for making figures [80].

### 4.9. Cloning and Purification of ∆N-Bcl-xL Recombinant Proteins

The rat ΔN-Bcl-xL cDNA was amplified by PCR using primers: (forward) 5′-GAAGGAGATACCACCATGgatagccccgcggtgaatgga-3′and (reverse) 5′-GGGCACGTCATACGGATActtccgactgaagagtgagcc-3′. PCR product was cloned directly into pME-HA vector using Expresso CMV Cloning and Expression System (Lucigen, Middleton, WI, USA), to introduce HA tags at the C terminus of ΔN-Bcl-xL. Next, HEK cells (ATCC, Manassas, VA) were transfected with pME-HA-ΔN-Bcl-xL plasmid, and cell lysates were prepared24 h later. ΔN-Bcl-xL protein was immunoprecipitated in the presence or absence of α-TCT using Pierce HA tag IP/Co-IP kit (Thermo Scientific, Rockford, IL, USA) according to the instruction manual. The samples were probed with anti-∆N-Bcl-xL and anti-Bax (monomeric) antibodies (Enzo, Farmingdale, NY).

### 4.10. Statistical Analysis

Data are reported as mean + SEM of at least three independent cultures with multiple independent experimental designs, such as independent neuronal isolation, independent performance dates, and independent plates within a culture using separately prepared reagents. All quantitative graphs were made from at least three independent neuronal isolation. Differences in means were tested using one-way ANOVA with Tukey’s test. *P* < 0.05 was considered statistically significant. *P* values are provided in figure legends.

## 5. Conclusions

This study shows a novel mechanism of α-TCT-mediated neuroprotection in primary hippocampal neurons. Treatment with α-TCT prevents ROS-induced ∆N-Bcl-xL formation, and α-TCT may bind directly to ∆N-Bcl-xL to inhibit the abundance and activity of ∆N-Bcl-xL. These actions block ∆N-Bcl-xL-Bax complex formation at the inner mitochondrial membrane, preventing mitochondrial dysfunction and mitochondrially mediated caspase-induced cell death.

## Figures and Tables

**Figure 1 ijms-21-00220-f001:**
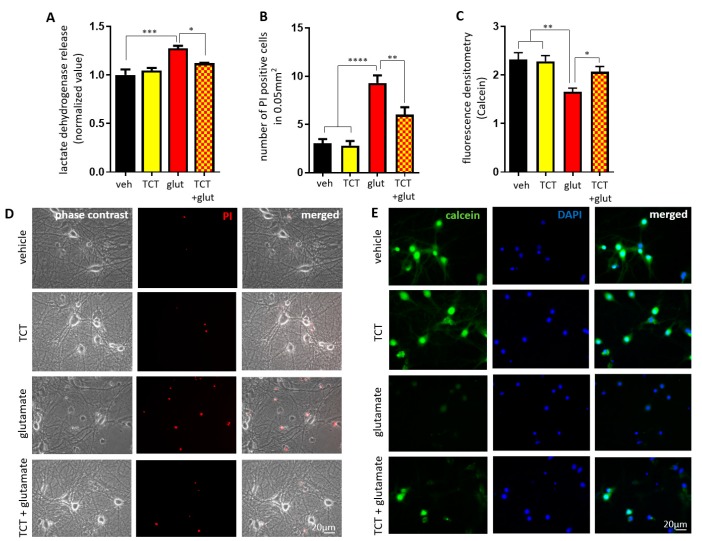
α-tocotrienol (TCT) protects against glutamate-induced death in primary hippocampal neurons. Primary hippocampal neurons were treated with α-TCT (1 µM), glutamate (20 µM), or a combination of both for 24 h. Quantified neuronal toxicity, death, and viability were measured by lactate dehydrogenase (LDH) release (*n* = 3 from three independent cultures) (**A**), PI positive cells (*n* = 20 micrographs per group) (**B**), and calcein retention (*n* = 35–39 micrographs per group) (**C**), respectively. PI-stained dead cells (**D**) or calcein-stained live cells (**E**) were imaged using a 32× fluorescent microscope. Hippocampal neurons treated with α-TCT were protected from glutamate-mediated death (Red: PI; green: calcein; blue: 4′,6-diamidino-2-phenylindole, DAPI). Scale bar = 20 µm. * *p* < 0.05, ** *p* < 0.01, *** *p* < 0.001, and **** *p* < 0.0001, one-way ANOVA.

**Figure 2 ijms-21-00220-f002:**
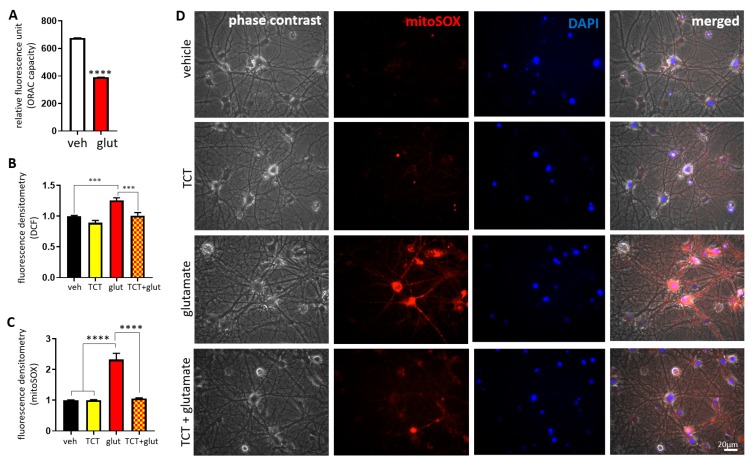
α-TCT attenuates glutamate-induced reactive oxygen species (ROS) production in the mitochondria. Primary hippocampal neurons were treated with α-TCT (1 µM), glutamate (20 µM), or a combination of both for 6 h. Quantification of intracellular lipophilic antioxidant capacity (**A**) and oxidative stress level (**B**) were assayed by measuring fluorescence intensity of fluorescein and 2′,7′-dichlorofluorescein (DCF) using the whole cell body (A, *n* = 6; B, *n* = 12), respectively. Mitochondrial oxidative stress levels were measured by mitoSOX staining. (**C**) Fluorescent intensity of mitoSOX (*n* = 15). (**D**) Glutamate treatment significantly increased fluorescence intensity of mitoSOX, whereas mitoSOX signal was attenuated by α-TCT co-treatment in primary hippocampal neurons (Red: mitoSOX; blue: DAPI). Scale bar = 20 µm. *** *p* < 0.001, and **** *p* < 0.0001, one-way ANOVA.

**Figure 3 ijms-21-00220-f003:**
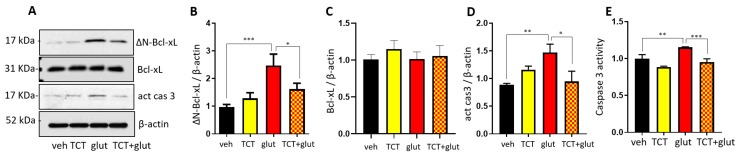
α-TCT prevents glutamate-induced cleaved B-cell lymphoma—extra large (ΔN-Bcl-xL) formation. Primary hippocampal neurons were treated with α-TCT (1 µM), glutamate (20 µM), or a combination of both. (**A**–**D**) Immunoblot data indicate that α-TCT treatment prevents glutamate-induced ΔN-Bcl-xL formation (**A**,**B**) without changing full length Bcl-xL protein levels (**A**,**C**). α-TCT also decreased the abundance of activated caspase 3 (**A**,**D**) measured at 24 h incubation (*n* = 4). (**E**) α-TCT treatment prevented glutamate-induced caspase 3 activity (*n* = 3) after 6 h incubation. * *p* < 0.05, ** *p* < 0.01, and *** *p* < 0.001, one-way ANOVA.

**Figure 4 ijms-21-00220-f004:**
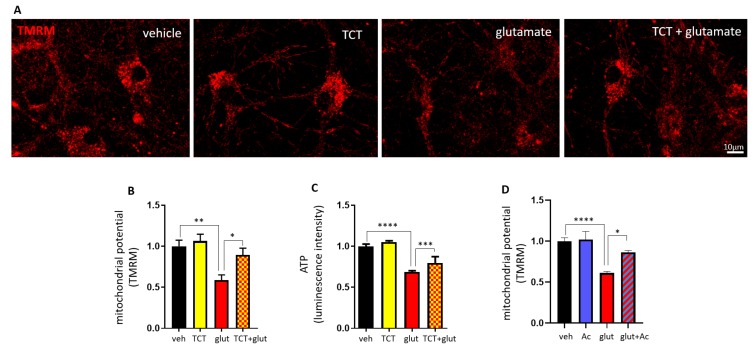
α-TCT protects mitochondrial function against glutamate challenge. Primary hippocampal neurons were treated with α-TCT (1 µM), glutamate (20 µM), or a combination of both. tetramethylrhodamine (TMRM)-stained neurons were imaged (**A**), and TMRM fluorescent intensity was quantified (**B**) at 6 h after treatment (*n* = 23–25 cells per group). (**C**) ATP levels (*n* = 24 wells per group) at 8 h after treatment. (**D**) Primary hippocampal neurons were treated with Ac-DEVD-CHO (10 μM), glutamate (20 µM), or a combination of both. TMRM fluorescent intensity was quantified (*n* = 20 cells per group). Scale bar = 10 µm. * *p* < 0.05, ** *p* < 0.01, *** *p* < 0.001, and **** *p* < 0.0001, one-way ANOVA.

**Figure 5 ijms-21-00220-f005:**
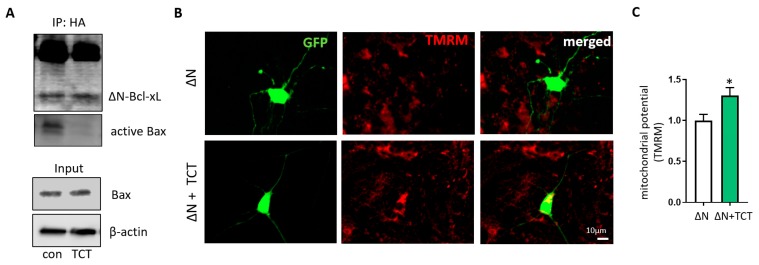
α-TCT protects mitochondrial function in ΔN-Bcl-xL-overexpressing neurons. (**A**), HEK293T cells were transfected with pME-HA-ΔN-Bcl-xL plasmid. ΔN-Bcl-xL protein was immunoprecipitated with or without α-TCT. ΔN-Bcl-xL protein immunoprecipitated in the presence of α-TCT showed decreased levels of active Bax interaction. Input lanes showed the presence of monomeric Bax in whole cell lysate. Primary hippocampal neurons expressing ΔN-Bcl-xL were treated with or without α-TCT. Neurons treated with α-TCT demonstrated attenuated ΔN-Bcl-xL-mediated loss of mitochondrial potential. TMRM-stained neurons were imaged (**B**), and TMRM fluorescence intensity was quantified (**C**) at 6 h after treatment (*n* = 9–10 cells per group). Scale bar = 10 µm. * *p* < 0.05, one-way ANOVA.

**Figure 6 ijms-21-00220-f006:**
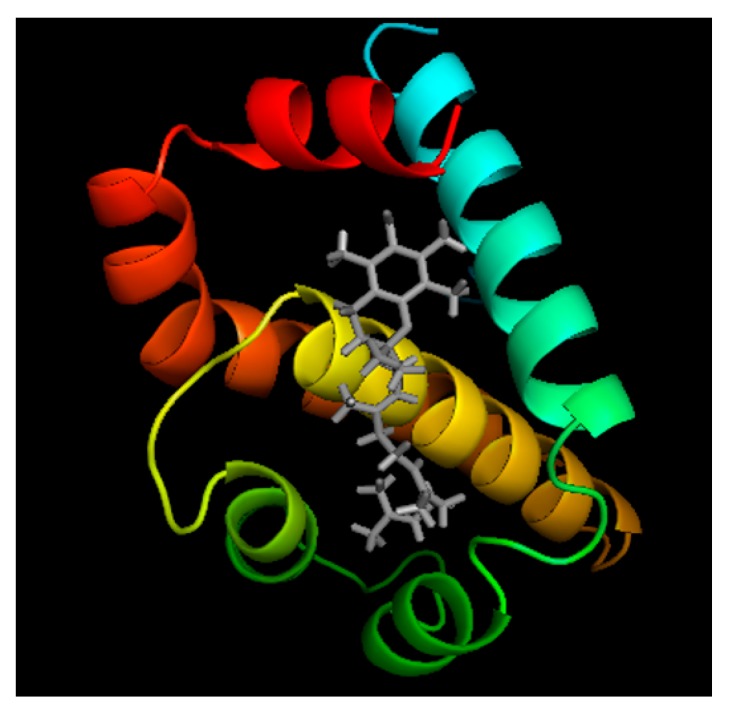
Model of α-TCT/ΔN-Bcl-xL complex. Docking of α-tocotrienol (grey) into the homology model of rat ΔN-Bcl-xL (Derived from PDB ID:1AF3).

**Figure 7 ijms-21-00220-f007:**
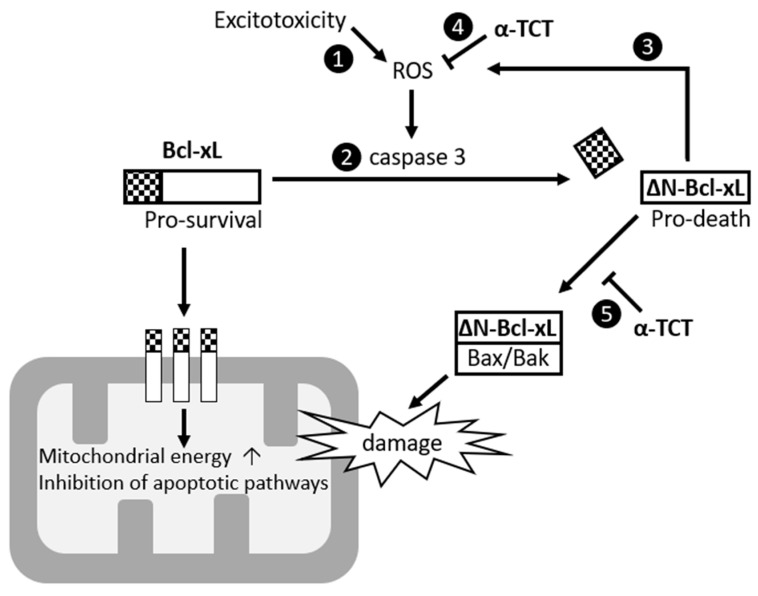
Schematic representation of ΔN-Bcl-xL-induced neuronal death and α-TCT mediated neuroprotection. Excess excitotoxic stimulation causes ROS production (1) in the mitochondria, which triggers caspase activation. Caspase 3 cleaves off the N-terminus of anti-apoptotic full length Bcl-xL to form pro-apoptotic ΔN-Bcl-xL (2). ΔN-Bcl-xL causes mitochondrial dysfunction and initiates mitochondrially mediated apoptotic signaling, which may result in a positive feedback loop to accelerate death pathways (3). This study showed that treatment with α-TCT prevents ΔN-Bcl-xL-mediated mitochondrial dysfunction by scavenging ROS (4), and we also suggest that α-TCT may occupy an active site of ΔN-Bcl-xL, preventing its access to other protein-binding partners such as Bax or Bak that regulate mitochondrial death pore opening (5).

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
