# Peer review of "Alpha-Tocotrienol Prevents Oxidative Stress-Mediated Post-Translational Cleavage of Bcl-xL in Primary Hippocampal Neurons"

_ijms, 2019, doi:10.3390/ijms21010220_

Round 1

Reviewer 1 Report

Park et al. study how α-Tocotrienol (α-TCT), an antioxidant, protects against neuronal cell death caused by glutamate treatment, which causes the generation of reactive oxygen species (ROS). A truncated form of Bcl-xL, ΔN-Bcl-xL, generated by cleavage by caspases during apoptosis, has been associated with mitochondrial dysfunction and neuronal cell death. The authors present evidence that glutamate treatment causes the accumulation of ΔN-Bcl-xL in primary neurons, and α-TCT suppresses the generation of glutamate-induced ΔN-Bcl-xL and neuronal cell death. Thus they suggest that ΔN-Bcl-xL is a target of α-TCT.

First, the authors did not provide evidence that ΔN-Bcl-xL is the cause or even part of the cause of glutamate-induced neurotoxicity in these cells. It is therefore questionable that ΔN-Bcl-xL is even relevant.

Second, there is no evidence that α-TCT has any physical or functional relationship with ΔN-Bcl-xL. α-TCT could have many relevant targets that help explain its neuroprotection.

Without answers to these questions, the study may not be significant.

Author Response

The authors appreciated suggestions and criticisms made about this manuscript. Please see the responses below.

Park et al. study how α-Tocotrienol (α-TCT), an antioxidant, protects against neuronal cell death caused by glutamate treatment, which causes the generation of reactive oxygen species (ROS). A truncated form of Bcl-xL, ΔN-Bcl-xL, generated by cleavage by caspases during apoptosis, has been associated with mitochondrial dysfunction and neuronal cell death. The authors present evidence that glutamate treatment causes the accumulation of ΔN-Bcl-xL in primary neurons, and α-TCT suppresses the generation of glutamate-induced ΔN-Bcl-xL and neuronal cell death. Thus they suggest that ΔN-Bcl-xL is a target of α-TCT.

First, the authors did not provide evidence that ΔN-Bcl-xL is the cause or even part of the cause of glutamate-induced neurotoxicity in these cells. It is therefore questionable that ΔN-Bcl-xL is even relevant.

We have published many papers elucidating mechanisms of ΔN-Bcl-xL-induced neurotoxicity using both in vivo and in vitro models (Jonas et al. 2004; Miyawaki et al. 2008; Ofengeim et al. 2012; Park and Jonas 2017; Park et al. 2017). We have added all references in the Introduction Section.

Second, there is no evidence that α-TCT has any physical or functional relationship with ΔN-Bcl-xL. α-TCT could have many relevant targets that help explain its neuroprotection.

α-TCT, is a nutrient. Therefore, yes.  We agree that α-TCT is involved in multiple pathways and we are not attempting to prove specificity of α-TCT. However, in this project, we would like to emphasize the alteration of intracellular redox homeostasis (please see new data added in Figure 2A) that is a significant event during excitotoxicity and likely to be a causative event in ΔN-Bcl-xL formation given that ΔN-Bcl-xL formation is blocked by α-TCT.

Reviewer 2 Report

This study is the continuation of previous works of the team regarding Bcl-xL roles on the bioenergetic status of nerve cells. The authors present an interesting set of preliminary data about the effect of alpha Tocotrienol (TCT) on glutamate-induced oxydative stress and the contribution of the pro-apoptotic deltaN-Bcl-xL truncated isoform in primary neurons.

Collectively the results support the notion that glutamate induces excessive mitochondrial ROS production, resulting in deltaN-Bcl-xL accumulation, drop in mitochondria delta psi, caspase activation and cell death.  They also provide some data indicating that TCT partially counteracts glutamate-induced cell death, by preventing ROS accumulation and caspase activation.

Major point :

a major gap in the study is that it is not known whether the activation of caspases is upstream or downstream of ROS accumulation. This should be checked e.g. using caspase inhibitors (see comments below). 
Specific points:

Figure 1 E. DAPI staining is almost undetectable. This should be corrected.

Figure 2 C. DAPI alone is missing. This should be corrected.

Figure 3. Is there any effect of glutamate on full length Bcl-xL level/subcellular distribution? This information should be added to panel A & E. Unfortunately, IHC pictures in panel E are not very informative due to poor resolution; subcellular distribution of deltaN-Bcl-xL should be adressed using alternative methods such as cell fractionation or transmission electron microscopy (immunogold).   

Figure 4. Overall glutamate seems to alter mitochondrial ETC. It is not clear if caspase activation is a cause or a consequence of mitochondrial metabolic dysfunction. This issue should be adress by using ZVAD-fmk or alternative caspase inhibitors.

Figure 5. Same comment as above. Any effect of TCT on caspase activity? Does caspase inactivation restores mitochondrial potential ?   

Author Response

The authors appreciated suggestions and criticisms made about this manuscript. Please see the responses below.

This study is the continuation of previous works of the team regarding Bcl-xL roles on the bioenergetic status of nerve cells. The authors present an interesting set of preliminary data about the effect of alpha Tocotrienol (TCT) on glutamate-induced oxydative stress and the contribution of the pro-apoptotic deltaN-Bcl-xL truncated isoform in primary neurons.

Collectively the results support the notion that glutamate induces excessive mitochondrial ROS production, resulting in deltaN-Bcl-xL accumulation, drop in mitochondria delta psi, caspase activation and cell death.  They also provide some data indicating that TCT partially counteracts glutamate-induced cell death, by preventing ROS accumulation and caspase activation.

 Major point :

a major gap in the study is that it is not known whether the activation of caspases is upstream or downstream of ROS accumulation. This should be checked e.g. using caspase inhibitors (see comments below). 

Specific points:

Figure 1 E. DAPI staining is almost undetectable. This should be corrected.

Changed

Figure 2 C. DAPI alone is missing. This should be corrected.

Changed

Figure 3. Is there any effect of glutamate on full length Bcl-xL level/subcellular distribution? This information should be added to panel A & E. Unfortunately, IHC pictures in panel E are not very informative due to poor resolution; subcellular distribution of deltaN-Bcl-xL should be adressed using alternative methods such as cell fractionation or transmission electron microscopy (immunogold).   

We have published that there is no significant change with Bcl-xL upon glutamate treatment, perhaps because of its high abundance compare to ΔN-Bcl-xL (Park at al. Cell Death and Differentiation, 2017). Figure 3E is removed.

Figure 4. Overall glutamate seems to alter mitochondrial ETC. It is not clear if caspase activation is a cause or a consequence of mitochondrial metabolic dysfunction. This issue should be adress by using ZVAD-fmk or alternative caspase inhibitors.

We have already reported data using various caspase inhibitors such as Ac-DEVD-CHO and zVAD-fmk previously (Jonas et al. 2004; Park et al. 2017). References are updated, and this information is more clearly explained in the text on page 137.

Figure 5. Same comment as above. Any effect of TCT on caspase activity? Does caspase inactivation restores mitochondrial potential?   

Again, we have already reported data using various caspase inhibitors previously (Jonas et al. 2004; Park et al. 2017). References are updated, and this information is more clearly explained in the text on page 137.

In addition, caspase is an upstream target of ΔN-Bcl-xL formation in this project, therefore application of caspase inhibitor where ΔN-Bcl-xL is already over-expressed (Figure 5) may not provide a direct mechanism.  ΔN-Bcl-xL will cause downstream ROS production with subsequent activation of caspase 3 and cleavage of full length Bcl-xL. This will be attenuated by TCT.

Round 2

Reviewer 1 Report

The authors adequately addressed the previous concerns in the revised manuscript.

Author Response

Thank you very much for your positive response.